# Actin Filament in the First Cell Cycle Contributes to the Determination of the Anteroposterior Axis in Ascidian Development

**DOI:** 10.3390/jdb10010010

**Published:** 2022-02-04

**Authors:** Toshiyuki Goto, Shuhei Torii, Aoi Kondo, Kazumasa Kanda, Junji Kawakami, Yosky Kataoka, Takahito Nishikata

**Affiliations:** 1Frontiers of Innovative Research in Science and Technology (FIRST), Konan University, 7-1-20 Minatojima-Minamimachi, Chuo-k, Kobe 650-0047, Japan; d1961001@s.konan-u.ac.jp (T.G.); m2161013@s.konan-u.ac.jp (S.T.); s1891017@s.konan-u.ac.jp (A.K.); kazu186607@gmail.com (K.K.); kawakami@konan-u.ac.jp (J.K.); 2Laboratory for Cellular Function Imaging, RIKEN Center for Biosystems Dynamics Research, Kobe 650-0047, Japan; kataokay@riken.jp; 3RIKEN-JEOL Collaboration Center, Multi-Modal Microstructure Analysis Unit, Kobe 650-0047, Japan

**Keywords:** axis determination, ER, actin, microtubule, maternal mRNA

## Abstract

In many animal species, the body axis is determined by the relocalization of maternal determinants, organelles, or unique cell populations in a cytoskeleton-dependent manner. In the ascidian first cell cycle, the myoplasm, including mitochondria, endoplasmic reticulum (ER), and maternal mRNAs, move to the future posterior side concomitantly (called ooplasmic segregation or cytoplasmic and cortical reorganization). This translocation consists of first and second phases depending on the actin and microtubule, respectively. However, the transition from first to second phase, that is, translocation of myoplasmic components from microfilaments to microtubules, has been poorly investigated. In this study, we analyzed the relationship between these cytoskeletons and myoplasmic components during the first cell cycle and their role in morphogenesis by inhibitor experiments. Owing to our improved visualization techniques, there was unexpected F-actin accumulation at the vegetal pole during this transition period. When this F-actin was depolymerized, the microtubule structure was strongly affected, the myoplasmic components, including maternal mRNA, were mislocalized, and the anteroposterior axis formation was disordered. These results suggested the importance of F-actin during the first cell cycle and the existence of interactions between microfilaments and microtubules, implying the enigmatic mechanism of ooplasmic segregation. Solving this mystery leads us to an improved understanding of ascidian early development.

## 1. Introduction

Maternal mRNA, which is produced from the maternal genome during oogenesis, is indispensable for body planning in many animal species. The localization and translation of maternal mRNA is strictly regulated during early embryogenesis. For the correct localization of maternal mRNA, cytoskeletons play a crucial role in the transport and anchoring of some animal eggs such as flog (cortical rotation) [1,2], teleost fish (cortical rotation) [3], and nematodes (cytoplasmic flow) [4,5].

Ascidian (Chordata) unfertilized eggs have a unique cytoplasm, designated as myoplasm, which consists of mitochondria-rich cytoplasm (MRC), cortical endoplasmic reticulum (cER), and maternal mRNAs called postplasmic/PEM RNAs [6,7]. In the first cell cycle, the myoplasm shows dynamic movement toward the future posterior side (called ooplasmic segregation or cytoplasmic and cortical reorganization) [8], and is important for anteroposterior axis formation [7].

This ooplasmic segregation consists of two phases. In the first phase, fertilization triggers the contraction of cortical actin filaments to the vegetal pole and the concentration of the myoplasm at the vegetal pole [9,10,11]. This first phase is completed within 5 min postfertilization (mpf). During the second phase, growing sperm aster migrates from the vegetal pole to the posterior side along the egg cortex, and finally, the sperm nucleus enters the center of the egg to fuse with the female nucleus [8]. The myoplasm moves toward the posterior pole following this sperm-aster movement and depends on the microtubules [9,10,11]. We recently reported the cortical array of microtubules in the posterior–vegetal region (CAMP) during the second phase, suggesting the importance of myoplasm translocation [12,13]. This second phase of ooplasmic segregation occurs roughly from 30 to 45 mpf. Although these two phases of ooplasmic segregations are well-described, the mechanisms of transition between the first and second phases are poorly understood. One of the concerns is how the myoplasm switches its chariot from F-actin to the microtubule.

Previously, cER and postplasmic/PEM mRNAs were thought to move posteriorly together during the second phase [7,8]. However, our recent observations showed that before the second phase, some postplasmic/PEM mRNAs dissociated from dense ER, which corresponds to the cER with a more deeply expanded ER mass [14]. Moreover, they reassembled by the four-cell stage and subsequently inherited in the centrosome attracting body (CAB) [14]. CAB is responsible for unequal cleavage and determines the anteroposterior axis [15,16]. These results strongly suggest the existence of unexpected mechanisms between the first and second phases of ooplasmic segregations and their importance for the anteroposterior axis formation.

In this study, owing to the improvement and refinement of the entire process of visualization techniques, we succeeded in revealing the prolonged F-actin accumulation at the vegetal pole up to 30 mpf. We analyzed the role of actin filaments during the first cell cycle on the segregation of myoplasm, including CAMP formation, translocation of macho-1, dense ER, and mitochondria and axis determination. The importance of the relationship between actin filaments and microtubules will be discussed.

## 2. Materials and Methods

### 2.1. Animal Experiments

Ascidian (*Ciona intestinalis* type A; also called *Ciona robusta*) adults were obtained from the National BioResource Project (NBRP), Tokyo, Japan. Methods for egg and sperm handling, fertilization, and dechorionation were performed as described previously [17,18]. The embryos were reared in filtered seawater at 18 °C. At this temperature, the first and second phase of ooplasmic segregations start immediately after fertilization and from approximately 30 min postfertilization (mpf), respectively. The first cleavage occurs at approximately 60 mpf. In the inhibitor experiments, eggs were treated with 2 µg/mL cytochalasin B (CytB; Sigma-Aldrich, St. Louis, MO, USA), 2.5 µg/mL nocodazole (Noco; Sigma-Aldrich), or the same dilution of solvent (0.1% DMSO) as a control during the desired periods. The eggs were exposed to these inhibitors with 3 mL seawater. Then, eggs were replaced to 10 mL seawater and washed 5 times with fresh seawater. It has been reported that cortical F-actin in CytB-treated culture cells could be recovered within 5 min after washout with this protocol [19].

### 2.2. Whole-Mount Immunofluorescent Staining

For microtubule staining, dechorionated *Ciona* eggs/embryos were fixed with 100% methanol at room temperature (approximately 25 °C) for 1 h. The fixed specimens were treated with ethanol up series (35%, 70%, and 100%) and stored at −30°C until further use. After washing with phosphate-buffered saline containing 0.05% Tween 20 (PBST), the specimens were treated with G1T0 (4 mol/L urea (MP Biomedicals, Santa Ana, CA, USA), and 1% glycerol in distilled water) for 90 min at 4 °C [12].

For double staining of the ER and microtubules, dechorionated *Ciona* eggs and embryos were fixed with Cold-Fix solution (3.2% formaldehyde in 80% methanol) at −30 °C for 1 h, followed by continued fixation at room temperature for 1 h with gentle shaking every 20 min. Both types of fixed specimens were treated with ethanol up series (35%, 70%, and 100%) and stored at −30 °C until further use. After washing with PBST, the specimens were treated with G1T0 for 90 min at 4 °C and then treated with antigen retrieval solution (modified from Hayashi et al., 2011; 6 M urea and 0.1 M Tris-HCl, pH 9.5) [20] for 30 min at 80 °C. The specimens were immunostained with the following antibodies: anti-α-tubulin mouse monoclonal antibody (anti-microtubule antibody; clone DM1A; Sigma-Aldrich; 1:100 dilution), anti-glucose-regulated protein 78 (GRP78; also known as Bip) rabbit polyclonal antibody (anti-ER antibody; StressMarq Biosciences, Victoria, BC, Canada; 1:100 dilution), Alexa Fluor 488-conjugated goat anti-mouse IgG antibody (Thermo Fisher Scientific; 1:1000 dilution), and Alexa Fluor Plus 555-conjugated goat anti-rabbit IgG antibody (Thermo Fisher Scientific; 1:1000 dilution). Nuclei were stained with 5 μg/mL 4′,6-diamidino-2-phenylindole dihydrochloride (DAPI). The stained specimens were then mounted with methyl salicylate (Nacalai Tesque, Kyoto, Japan).

### 2.3. Phalloidin Staining

For F-actin staining, dechorionated *Ciona* eggs/embryos were fixed with a 1:1 mixed solution of extraction buffer (2% Triton-X 100, 50 mM MgCl_2_, 10 mM KCl, 10 mM EGTA, 20% glycerol, 25 mM imidazole) and fixation buffer (0.25% glutaraldehyde (Nacalai Tesque), 3.7% formaldehyde, 100 mM HEPES (pH = 7.0), 50 mM EGTA, 10 mM MgSO_4_, and 525 mM sucrose) [21] for 15 min at room temperature. This was followed by continued fixation in fixation buffer for 2 h at room temperature. The fixed specimens were washed with PBST and treated with 10 unit/mL Alexa Fluor 488-labeled phalloidin (Molecular Probes, Eugene, OR, USA) for 30 min at room temperature. For clearing without dehydration, stained specimens were treated with 40% fructose (Fujifilm-Wako, Tokyo, Japan) for 30 min at room temperature and then mounted in SeeDB (80.2% wt/wt fructose, 0.5% α-thioglycerol) [22].

### 2.4. Whole-Mount RNA In Situ Hybridization

Dechorionated *Ciona* eggs/embryos were fixed with a Cold-Fix solution. Fixed specimens were treated with 0.2% Triton-X 100 for 10 min at room temperature and then fixed with paraformaldehyde for 1 h at room temperature. The postfixed specimens were treated with 0.1 M 2,2′,2′-nitrilotriethanol (Fujifilm-Wako) and 0.27% acetic anhydride (Nacalai Tesque) for 10 min at room temperature. Then, specimens were treated with G1T0 followed by the mixture of antigen retrieval solution and prehybridization solution containing 3.78 M urea, 0.063 M Tris-HCl (pH = 9.5), 50 µg/mL heparin, 100 µg/mL yeast tRNA, and 1% Tween 20. Sense and antisense RNA probes of macho-1 were transcribed from a *Ciona* cDNA clone (cieg016n12: Ghost) [23], using T7 and T3 RNA polymerases (Sigma-Aldrich), respectively, with DIG RNA Labeling Mix (Sigma-Aldrich). The specimens were hybridized with an RNA probe in prehybridization buffer (50% formamide, 50 µg/mL heparin, 100 µg/mL yeast tRNA, and 1% Tween 20) for 16 h at 50 °C. After hybridization, the specimens were washed with SSC (5×, 2×, and 0.2×). For observation of macho-1, DIG probe was detected using an alkaline phosphatase-conjugated anti-DIG Fab fragment (Sigma-Aldrich; 1:1000 dilution). In contrast, when combined with immunofluorescence, specimens were immunostained with the following antibodies: anti-Bip rabbit polyclonal antibody, anti-NN18 mouse monoclonal antibody (antineurofilament antibody; Sigma-Aldrich; 1:100 dilution), horseradish peroxidase-conjugated anti-DIG Fab fragment (Sigma-Aldrich; 1:100 dilution), Alexa Fluor Plus 555-conjugated goat anti-rabbit IgG antibody, and Alexa Fluor 405-conjugated goat anti-mouse IgG antibody (Thermo Fisher Scientific; 1:200 dilution). The NN18 antibody is a good marker of mitochondria that recognizes the F1-ATP synthase α-subunit in *Ciona* [24]. After immunostaining, the in situ hybridization signals were enhanced using FITC-Tyramide (Akoya Biosciences, Marlborough, MA, USA).

### 2.5. Image Acquisition and Data Analysis

Specimens were observed under LSM700 (Carl Zeiss, Jene, Germany) or A1RHD25 (Nikon, Tokyo, Japan) confocal microscopes using ZEN (Carl Zeiss) or NIS element imaging software (Nikon). Z-projections and the midplane optical sections are displayed with animal pole upward. The animal pole was defined based on the position of the meiotic apparatus, myoplasm configuration, or parallel direction of the CAMP [12]. All analyses were performed using the ImageJ software [25]. Z-projections were displayed as a side view wherein the animal pole was up and posterior side was right, or as a posterior view wherein the animal pole was up. These Z-projections showed the maximum intensity projection of z-stack. To analyze the expansion of the signal area of macho-1 mRNA, the angles between the horizontal and vertical edges of the mRNA signal and the center of the eggs were measured. For the measurement of angles, the side and top views of the Z-projections were used. To analyze the colocalization between the ER and mRNA signals, Z-projections were rendered from five optical sections around the midplane. Then, the dense ER region was extracted using denoising, contrast adjustment, and binarization, followed by a restriction of the object size. The maternal mRNA-positive region was extracted by binarization of the mRNA signals. The ratio of the mRNA-positive regions in the dense ER to the total area of mRNA-positive regions was calculated. The parameters for the extraction of each dense ER or mRNA were fixed for all specimens.

## 3. Results

### 3.1. Prolonged F-Actin Accumulation at the Vegetal Cortex

It has been well-described that the actin cytoskeleton is necessary for the first phase of ooplasmic segregation and results in strong F-actin accumulation in the vegetal cortex [9,10,11]. However, no report has described how long this accumulation lingered in the vegetal pole. Thus, we improved the phalloidin staining method for detecting F-actin in whole-mount specimens. The main points of this improvement were fixation with permeabilization and treatment with the fructose-based hydrophilic clearing reagent SeeDB [22]. Our methods revealed that vegetal accumulation of F-actin decreased gradually, though it was detected up to at least 30 mpf (Figure 1). The mitotic apparatus was faintly stained at 45 mpf (Figure 1b). As the meiotic spindles beneath the animal cap were not stained, this phalloidin staining of the microtubule structure suggests that the colocalization of actin and tubulin during the second phase of ooplasmic segregation is plausible.

### 3.2. Actin Depolymerization Induced Malformation of CAMP

To reveal the functions of prolonged F-actin accumulation at the vegetal cortex during the following events, eggs were treated with CytB at various periods during the first cell cycle; CytB_01 (0 to 10 mpf), CytB_02 (0 to 30 mpf), CytB_03 (10 to 30 mpf), and CytB_04 (30 to 45 mpf), as shown in Figure 2a. To investigate the effect of CytB treatment on axis formation, we analyzed the second phase of ooplasmic segregation extensively. The effect of CytB treatment was confirmed by phalloidin staining of normal eggs fixed at 10 mpf. In those eggs, no F-actin staining in the vegetal pole could be found (Figure 2b,b′). First, microtubule structures in the eggs of 45 mpf were observed. In the control egg, the CAMP was normally formed, representing a kind of convergence to the midline; thus, microtubule bundles were dense around the midline (Figure 2c). The CAMP formed by CytB_01 and CytB_03 treatments were relatively normal, though expanded broadly, and did not show midline accumulation of microtubule bundles (Figure 2d,f). The CytB_02 treatment gave rise to a widely and randomly distributed microtubule network in the vegetal hemisphere (Figure 2e). In the CytB_04 treatment, although the microtubule bundles were arrayed in a similar direction and accumulated around the postulated posterior pole, this CAMP-like structure was missing a large portion (Figure 2g). We supposed that the treatment with CytB before the second phase of ooplasmic segregation broadened the CAMP-forming area and made the anteroposterior axis ambiguous, whereas the treatment during the second phase of movement caused a defective CAMP with the postulated anteroposterior axis. Moreover, although several chromosomes detached from spindle microtubules in CytB_02 (Figure 2j), mitotic apparatus for the first cleavage had bipolar spindles and asters in all CytB-treated eggs (Figure 2h–l). Thus, the effect of CytB treatment mainly affected the microtubules beneath the egg cortex, suggesting a relationship between microfilaments and microtubules beneath the egg cortex.

### 3.3. The Prolonged F-Actin Localization Contributed to the Midline Accumulation of Myoplasm and Maternal mRNA

As the effect of CytB on CAMP formation was different, we first focused on the effect before the second phase of ooplasmic segregation (0 to 30 mpf). The CytB-treated eggs were fixed at 45 mpf and triple-stained for macho-1 mRNA, ER, and mitochondria (Figure 3). In DMSO-treated control embryos, dense ER, MRC, and macho-1 were clearly localized and concentrated at the posterior pole, and both mitochondria and macho-1 were excluded from the dense ER region, as reported in our previous paper (Figure 3a,a′,h). When eggs were treated with CytB, the distribution of macho-1 broadly expanded with various phenotypes (Figure 3b–d,b′–d′,b″–d″). In CytB_01, although a large part of macho-1 was localized to the posterior side, some macho-1 were left in anterior side (Figure 3b,b′). Moreover, the localized area of macho-1 was broadened along a left–right axis (Figure 3b″). This result might be due to the retention of macho-1 on almost the entire egg cortex caused by the inhibition of the first phase of ooplasmic segregation. In CytB_02, macho-1 was broadly localized in the cortical regions of the vegetal hemisphere (Figure 3c,c′,c″). This severe phenotype was thought to be the additive effect of CytB_01 and CytB_03. In most cases of CytB_03, macho-1 moved and localized to the posterior pole, similar to those of the normal control (Figure 3d,d′), however, it showed wider distribution along the left–right axis (Figure 3d″). Notably, CytB_01 and 02 treatments showed scattered distributions of macho-1 signals, while CytB_03 treatment showed a single lump of macho-1 signal (Figure 3b″–d″). In the quantitative analyses of these results, it was clearly demonstrated that CytB treatments had different effects for the macho-1 distributions along the anteroposterior and left–right axes (Figure 3e–g). To summarize these results, depolymerization of F-actin during the first segregation movement resulted in the scattered and broadened localization of macho-1 along both anteroposterior and left–right axes, while the depolymerization of F-actin during the period between the first and second segregation only broadened the macho-1 localization along the left–right axis. Thus, it was suggested that the actin filaments during the first phase of segregation had the role of lumping macho-1 mRNAs into a single mass in addition to the constriction of the egg cortex into the vegetal pole, and prolonged localization of actin filaments beneath the vegetal pole had the role of the convergence of macho-1 mRNAs to the midline. The MRC resided adjacent to the inner side of dense ER, and macho-1 extruded from dense ER and into the MRC and deeper cytoplasm in all treatments (Figure 3h–k). Although, the macho-1 mRNA was distributed abnormally by CytB treatments, dense ER and MRC were situated close to macho-1, and MRC was always situated next to the dense ER.

### 3.4. The Role of F-Actin during the Second Phase of Ooplasmic Segregation

Eggs treated with CytB_04 were fixed at 45 mpf and triple-stained for ER, mitochondria, and macho-1 (Figure 4). In normal eggs, dense ER thickened its localized area at the posterior pole region and tended to intrude into the inner cytoplasm by following the movement of sperm aster, and macho-1 was excluded from the ER and intruded into the MRC or even deeper cytoplasm of the posterior pole (Figure 4a,a′,c,c′). In the CytB_04-treated egg, both macho-1 and dense ER were situated on the posterior vegetal cortex (Figure 4b,b′). However, they formed a thin localized area sticking to the cortex, and most of the macho-1 signal colocalized with dense ER (Figure 4d,d′). To examine the role of microtubules in the macho-1 translocation during the second phase of ooplasmic segregation, eggs were treated with 2.5 µg/mL Noco from 30 to 45 mpf (Figure 4e–e″,f–f‴). Noco treatment inhibited the second phase of ooplasmic segregation. Thus, the myoplasm remained in the vegetal pole, and the anteroposterior axis was abolished. Although no microtubule staining was observed in Noco-treated eggs, there was a faint tubulin staining at the vegetal pole (Figure 4e,e′). Dense ER colocalized with this tubulin-staining region, suggesting a close relationship between the ER and microtubules (Figure 4e″). According to the triple-staining result, mutual positioning of the vegetally stained ER and MRC was basically the same as the normal positioning of 30 mpf (Figure 4f,f′) and of the CytB_04 treatment. The MRC resided adjacent to the inner side of dense ER (Figure 4f‴). Although dense ER showed a thin layer beneath the posterior cortex and did not move into the egg cytoplasm in CytB_04 treatment, Noco treatment did not change dense ER and MRC distribution (Figure 4d,d′,f,f′,f‴). On the other hand, in Noco treatment, most of the macho-1 signals colocalized with dense ER, similarly to CytB_04 (Figure 4d,d′,f,f′,f″). Quantitative analysis of this abnormality in the translocation of macho-1 revealed a similar effect of both actin and tubulin depolymerizations (Figure 4g,h). As macho-1 dissociated from dense ER by 30 mpf in the normal condition [14], macho-1 went back to dense ER under the condition with each inhibitor. Although, there is a possibility that the depolymerization of microtubules indirectly affects the macho-1 translocation by aborting the movement of the second phase, both F-actin and microtubules were suggested to have some roles in preventing macho-1 being reassociated with dense ER during the second phase of movement.

### 3.5. F-Actin Contributed to the Cleavage Patterning during the First Cell Cycle

The effect of CytB treatment during the first cell cycle to the later cleavage was investigated, focusing on the establishment of the anteroposterior axis. The eggs treated with CytB were fixed at the 32-cell stage and stained for macho-1 (Figure 5). In normal 32-cell stage embryos, macho-1 was localized to the CAB region, which resides equally in the posterior-most vegetal small blastomere pair (B6.3 and B6.3; micromere), and defined the posterior pole (Figure 5a). When the eggs were treated with CytB at various periods of the one-cell stage, they showed various abnormal phenotypes. Two-thirds of the embryos of CytB_01 treatment gave rise to the embryo with two macho-1-localized spots in the adjacent micromeres, predicting the establishment of the anteroposterior axis, but the sizes of the signal areas and micromeres were unequal (Figure 5b). On the other hand, one-third of CytB_01-treated embryos had two macho-1 localized spots in two discontinuous blastomeres, which could not define their posterior pole (Figure 5b′). CytB_02-treated embryos showed the most severe phenotype. Most of the embryos showed delayed and abnormal cleavage with no symmetrical pattern, even in the embryo with two macho-1 signals in neighboring blastomeres (Figure 5c,c′). Three-fourths of them had a single elongated signal situated unevenly in two blastomeres (Figure 5c′). The majority of the CytB_03-treated embryos had two or three macho-1-localized spots in the equivalent-size blastomeres, and thus could not show the anteroposterior axis (Figure 5d,d′). However, half of the CytB_04-treated embryos had two macho-1-localized spots with weak left–right symmetry. The other half had only one macho-1-localized spot with no left–right symmetry (Figure 5e,e′).

These embryos were then stained for macho-1 mRNA, ER, and mitochondria to observe their relative locations (Figure 6). In normal 32-cell stage embryos, macho-1 mRNA colocalized with ER in the micromeres corresponding to CAB formation (Figure 6a,a′). Although CytB-treated embryos showed an abnormal distribution pattern of macho-1 after the second phase of ooplasmic segregation, localized spots of macho-1 in all the treatments were colocalized with ER, suggesting that CAB formation normally occurred (Figure 6b–g,b′–g′). As shown in Figure 5, some of the CABs did not lead to unequal cleavage and posterior pole formation. These CABs suggested a lack of some factors, which is inevitable for exerting the complete function of CAB. Our results suggested that depolymerization of F-actin during the first cell cycle affects the formation of normal CAB.

### 3.6. The Disordered Morphogenesis Induced by Cytb Treatment during First-Cell Cycle

Finally, to examine the effect of CytB treatment during the first cell cycle on normal CAB formation, the larval phenotype was examined (Figure 7). Normal larva represented a tadpole shape with a trunk and long tail representing obvious anteroposterior axis (Figure 7a). All of the CytB treatments led to abnormal development and gave rise to two phenotypes (Figure 7b–e,b′–e′). In one phenotype, irregular trunk and abnormal short tail could be identified, suggesting anteroposterior axis formation. In some cases, a pigment granule (thought to be an otolith according to its round shape) was developed (Figure 7b,e). This type of larvae developed approximately 60%, 25%, 45%, and 55% in CytB_01, CytB_02, CytB_03, and CytB_04, respectively (Figure 7b′–e′). These proportions were close to those of the 32-cell stage embryo with left–right symmetry, as shown in Figure 5. In the other phenotype, although the secretion of tunic suggesting epidermal development and faint rotation symmetry could be observed, it was a type of cell aggregate and we could not define the anteroposterior axis (Figure 7b′–e′). These embryos were thought to be develop from the 32-cell-stage embryo with no anteroposterior axis (Figure 5). This result implied that some CABs could not exert complete functions nor define the anteroposterior axis, suggesting that some important factors were missing from CAB by CytB treatment.

## 4. Discussion

Owing to the methods using unique fixation and hydrophilic clearing reagents, we found prolonged actin localization to the vegetal pole until the start of the second phase. Moreover, we analyzed the role of F-actin during the first cell cycle by describing the colocalization of ER, mitochondria, and mRNA.

When F-actin was depolymerized by CytB at various periods, the cortical microtubule structure, CAMP, showed abnormal shapes according to the period of CytB treatment. The common phenotype of the depolymerization of prolonged vegetal F-actin was the laterally broadened CAMP area, suggesting the inhibition of the convergence of microtubule bundles to the midline. The function and localization of F-actin during the first cell cycle of ascidian early development were reported mainly on the first phase of ooplasmic segregation [9,10,11], except for a few reports on the existence of cortical F-actin in the posterior–vegetal cortex during the second phase [27], which we could not confirm in our experiment. Thus, this is the first report that F-actin has some roles, in addition to the first phase of ooplasmic segregation. Specifically, in this report, we found three different roles for maternal mRNA translocation in three different periods, during the first phase of movement, the period between first and second phase, and the second phase of movement. During the first phase, a cortically distributed actin network was suggested to have a role for lumping maternal mRNA in addition to generating contracting force of the first segregation movement. During the period between first and second segregation, prolonged vegetal cortex F-actin was closely located to the vegetally accumulated microtubule fragment [13] and had a role in the convergence of mRNA, mitochondria, and ER to the midline. During the second phase, although we could not detect cortical actin network, depolymerization of F-actin affected the CAMP formation and relocalization of maternal mRNA to the dense ER region. The role for keeping maternal mRNAs dissociated from ER was shared by microtubules. These close relationships between microfilaments and microtubules strongly suggest their collaborative mechanisms.

The collaborative role of actin and tubulin in CAMP formation is not an inconceivable event, as it has been reported that actin filaments in somatic cells have various functions in microtubule organization, such as anchoring to the cortex and directional elongation [28]. Moreover, CAMP is suggested to be an acentrosomal microtubule structure. It has been reported that the acentrosomal microtubule structure could be regulated by actin through the spectraplakin (ACF7; crosslinking cytoskeletal protein by binding to both microtubules and F-actin) and the microtubule end-binding proteins (CAMSAP3: calmodulin-regulated spectrin-associated protein 3) [29]. Thus, to reveal the interaction between actin and microtubule in the ascidian egg, analyses of these cytoskeletal-associated molecules are necessary.

The macho-1 dissociation from dense ER during the second phase of ooplasmic segregation was another collaborative mechanism of actin and tubulin. It is well-known that both microtubules and microfilaments have a role in trafficking mRNP granules, complexes of mRNA and ribonucleoproteins (e.g., [30]). In budding yeast, it has been reported that the localization of polysome-interacting protein on the ER is regulated by microtubules [31]. In *Ciona* eggs, one of the RNA binding proteins, Y-box binding protein (YB-1), interacts with *pem-1* mRNA, suggesting the existence of mRNP granule [32]. Type I postplasmic/PEM RNAs associate with ribosomes, which are localized to the cER [33]. From the viewpoint of the collaborative role of different cytoskeletal filaments, an intangible molecular mechanism emerges by combining these reports. In this mechanism, translocation of organelles and mRNA are closely related to translational control. In our results, the importance of excluding mRNA from dense ER is cryptic, whereas the collaborative participation of two different cytoskeletal filaments is a promising mechanism for understanding the translocation of maternal determinants.

On the other hand, the malformation of microtubule structure and the mislocalization of mRNA caused by the depolymerization of microfilaments during the first cell cycle largely affected early development by spoiling the anteroposterior axis. Although most of these damaged embryos had CABs, they could not exert their complete functions, such as determination of the posterior pole and unequal cleavage. This suggested that the components of CABs formed in damaged embryos were insufficient. A considerable number of molecules have been revealed, such as about 40 postplasmic/PEM RNAs and their translational products, Kinesin, β-catenin, YB-1, and regulators of translation initiation [7,16,32,34,35,36,37]. Thus, more detailed analyses including these CAB components are required for understanding the relationship between dysfunction of CAB and incomplete translocation of myoplasm. It is well-known that the correct transport of myoplasm and the appropriate distribution of the myoplasmic components through the ooplasmic segregation are important for ascidian development. According to our results, we believe that the precise mechanisms of ooplasmic segregation are more complicated than ever thought. The collaborative role of microfilaments and microtubules is one of these complexities. More extensive analyses of the mechanisms underlying ooplasmic segregation will contribute to the understanding of the molecular mechanisms of ascidian embryogenesis.

## Figures and Tables

**Figure 1 jdb-10-00010-f001:**
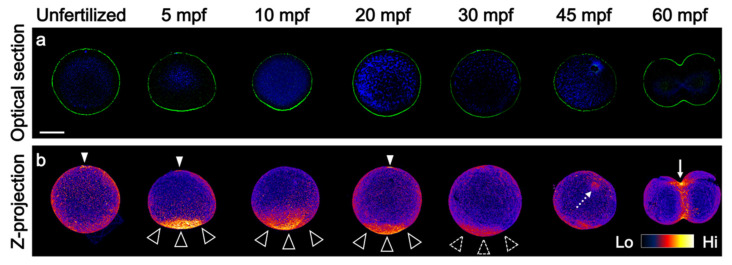
Duration of vegetally localized F-actin after fertilization. (**a**) The embryos during first cell cycle were stained for DNA (blue) and F-actin (green) by using 4′,6-diamidino-2-phenylindole dihydrochloride (DAPI) and phalloidin. The optical sections of the midplane are shown except for 60 mpf. At 60 mpf, the frontal view wherein the animal pole is up and right side is right was shown, because the cleavage furrow can be easily observed in this plane. The timings of fixation are indicated on top. The F-actin significantly localized to the vegetal pole at least until 30 mpf. Closed arrowheads in 5 and 20 mpf indicate actin caps of polar body I and II, respectively, suggesting the existence of meiotic spindle beneath them [26]. These observations were repeated four times in each time point using approximately 50 eggs in each experiment. (**b**) Z-projection models were rendered from single channel of F-actin and represented in fire gradient. The range indicator is shown at the right corner. Arrowheads and dotted arrowheads indicate the strong and weak signals of F-actin at the vegetal pole, respectively. The staining of the other cortical region including 45 mpf was not obvious nor consistent. Mitotic apparatus was slightly stained in 45 mpf (dotted arrow) and contractile ring of cleavage furrow was brightly stained in 60 mpf (arrow). Animal pole (A) is up. Scale bar: 50 µm.

**Figure 2 jdb-10-00010-f002:**
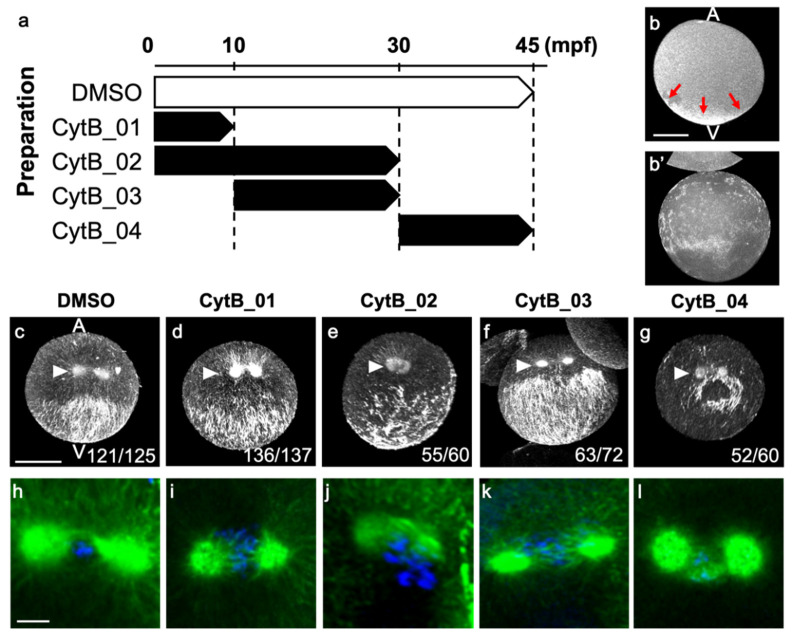
The effect of CytB treatment on CAMP formation during first cell cycle. (**a**) Schematic drawing of sample preparation. The white and black blocks indicate incubation periods with DMSO and 2 µg/mL CytB, respectively. The egg was treated with DMSO as the normal control. (**b**,**b′**) Normal (**b**) and CytB-treated eggs (**b′**) were fixed at 10 mpf and stained with phalloidin. CytB-treated eggs did not show F-actin staining in the vegetal pole. The Z-projections of the side view are shown. Animal pole (A) is up. Arrows indicate strong signal at the vegetal pole. Scale bar: 50 µm. (**c**–**g**) Eggs were fixed at 45 mpf and immunostained for tubulin. Preparation name of each egg is indicated on the top. Animal pole (A) is up. Posterior views of rendered Z-projection representing various malformations of CAMP. Normal CAMP emerges as a parallel array of microtubules on the posterior-vegetal cortex. Microtubule bundles are relatively dense around the midline (**c**). Arrowhead indicates mitotic apparatus of first cleavage, representing the midplane is situated in between two centrosomes. It should be noted that, although the progression of the cell cycle was slightly affected (**d**). The number of embryos represented by the image over the number of embryos examined is indicated at the right corner. Scale bar: 50 µm. (**h**–**l**) Enlarged Z-projections rendered from 30 optical sections around the mitotic apparatus in (**c**–**g**) are represented. Green and blue indicate microtubule and DNA, respectively. Scale bar: 10 µm.

**Figure 3 jdb-10-00010-f003:**
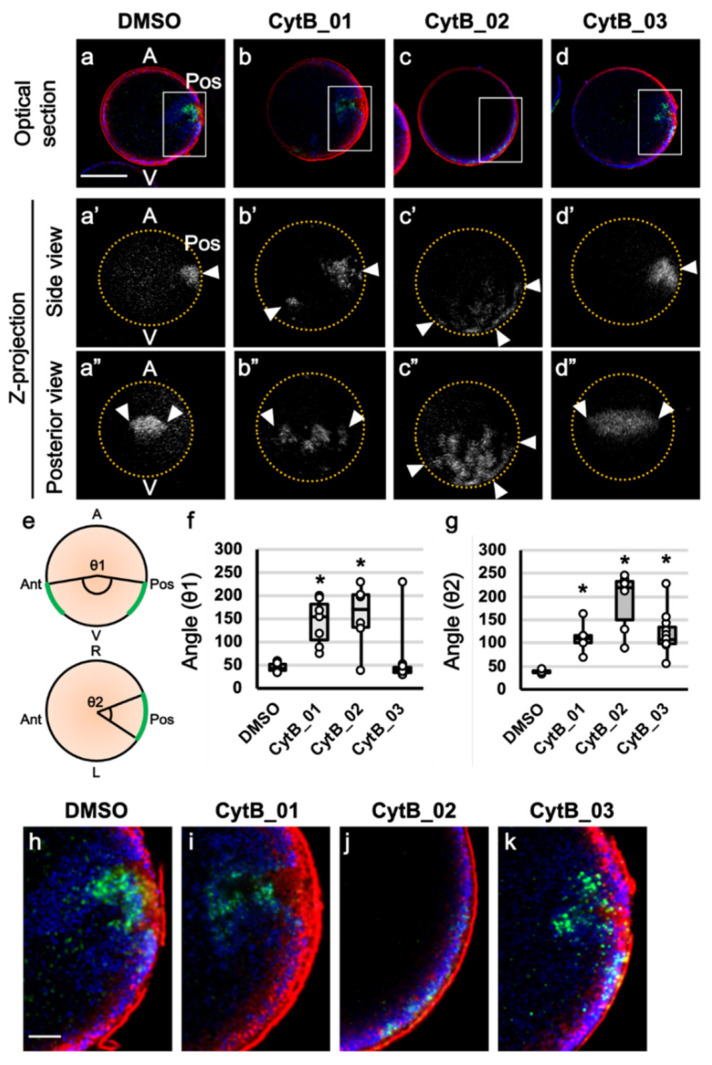
The role of prolonged F-actin localization to the vegetal cortex for translocations of organelles and maternal mRNA. (**a**–**d**,**a′**–**d′**,**a″**–**d″**) The CytB-treated eggs were fixed at 45 mpf. Preparation of each egg is indicated on the top. The double-immunostaining of ER (red) and mitochondria (blue) and in situ hybridization of macho-1 (green) were carried out. Merged images of midplane optical sections ((**a**–**d**): optical section) and Z-projections of single channel of macho-1 are shown ((**a′**–**d′**): side view, (**a″**–**d″**): posterior view). Yellow dotted lines indicate the outline of eggs. Animal pole (A) is up, vegetal pole (V) is down, and posterior pole (Pos) is on the right. Arrowheads indicate mRNA localized region. Scale bar: 50 µm. (**e**) The angles, θ1 and θ2, were measured between vertical and horizontal edges of mRNA signal area, respectively, and the center of eggs using Z-projection as shown in the schema. In CytB_01, macho-1 signals were left in the anterior side, and thus, θ1 became relatively large in most cases. (**f,g**) The central angle of the macho-1-localized region was measured on Z-projection and represented using box plots. Numbers of specimens were 7, 7, 6, and 11 in DMSO, CytB_01, _02, and _03, respectively. These angles represent expansion of macho-1 localization with CytB treatments. Statistical significance was calculated by one-way ANOVA followed by the Dunnett’s test (SD in f: DMSO = 9.6, CytB_01 = 50.2, CytB_02 = 69.7, CytB_03 = 58.4; SD in g: DMSO = 3.8, CytB_01 = 30.9, CytB_02 = 64.9, CytB_03 = 43.7). Significant differences versus DMSO are represented by asterisks (*p* < 0.01). (**h**–**k**) Enlarged images of white rectangles of (**a**–**d**) are represented. In DMSO and CytB_01, MRC (blue) was adjacent to the inner side of dense ER (red), while in CytB_02 and _03, MRC and dense ER were mingled but represented discrete distribution (blue and red signals do not overlapped). Scale bar: 10 µm.

**Figure 4 jdb-10-00010-f004:**
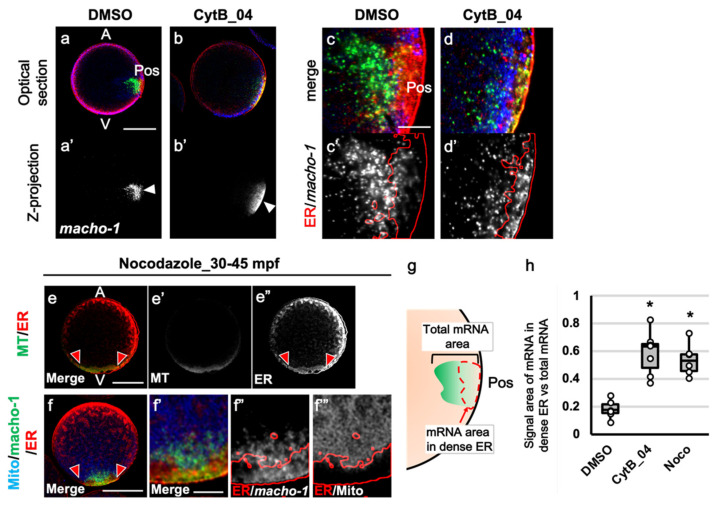
The role of F-actin during second phase of movement for the translocation of organelles and maternal mRNA. (**a**–**d**,**a′**–**d′**) The CytB-treated eggs were fixed at 45 mpf and double-immunostaining of ER (red) and mitochondria (blue) and in situ hybridization of macho-1 (green) were carried out. Preparation of each egg is indicated on the top. Merged images of midplane optical sections ((**a,b**): optical section) and rendered Z-projections of single channel of macho-1 signal ((**a′**,**b′**): Z-projection) are indicated. (**c**,**d**) Enlarged merged images of the posterior pole regions of a and b are shown (Merge). (**c′**,**d′**) The outline of dense ER region (red line) was superimposed on single channel of macho-1 signal of c and d (white). Animal pole (A) is up, vegetal pole (V) is down, and posterior pole (Pos) is on the right. Arrowhead indicates *macho-1*-localized region. (**e**–**e″**,**f**–**f″**) Noco treatment was carried out during 30–45 mpf and fixed at 45 mpf. Single optical section of the plane including animal–vegetal axis are shown. Regions between two red arrowheads represent dense ER region. Noco-treated eggs were double stained for ER (red) and tubulin (MT: green). Merged images ((**e**): Merge) and single channel of each signal ((**e′**); MT, (**e″**); ER) are represented. Microtubule staining was not present, while faint staining was observed in vegetal pole region, presumably due to the tubulin monomer staining (**e′**). This tubulin staining was colocalized with dense ER (**e**,**e″**). Nocodazole-treated eggs were triple stained for ER (red)/mitochondria-rich cytoplasm (MRC; blue)/macho-1 (green). Merged images (**f**: Merge) and enlarged image of vegetal pole region of f ((**f′**); Merge) are shown. Outlines of the dense ER region (red lines) were superimposed on in situ hybridization signals ((**f″**); ER/macho-1) and MRC signals ((**f‴**); ER/MRC). Note that the background ER staining of entire cortex of the egg became blotchy. Scale bars: 50 µm (**a**,**e**,**f**) and 10 µm (**c**,**f′**). (**g**) Colocalization between macho-1 and dense ER was quantitatively evaluated by calculating the ratio of the signal area of the macho-1 within the dense ER region to the total area of mRNAs, as shown in the schema. (**h**) The results of ratio of the macho-1 signal area are represented using box plots. Dissociation of macho-1 from dense ER was inhibited by both CytB_04 and Noco treatments. The number of specimens was 6 in all treatments. Statistical significance was calculated by one-way ANOVA followed by the Dunnett’s test (SD: DMSO = 0.07, CytB_04 = 0.15, Noco = 0.11). Significant differences versus DMSO are represented by asterisks (*p* < 0.01).

**Figure 5 jdb-10-00010-f005:**
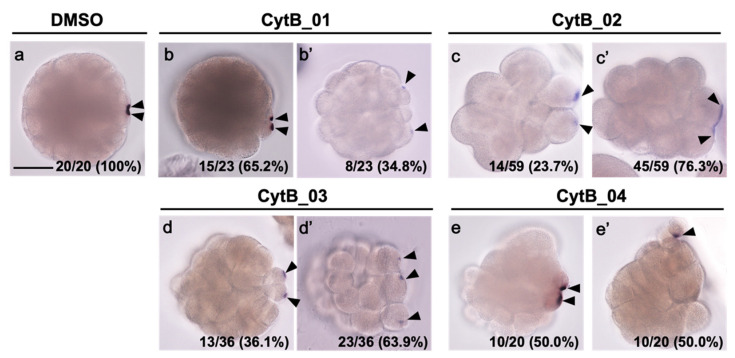
The effect of CytB treatment during 1-cell stage on the macho-1 mRNA localization. Preparation name of each egg is indicated on the top. Embryos were fixed at the 32-cell stage (about 3 h after fertilization) and stained for macho-1 by in situ hybridization. (**a**) The embryo treated with DMSO during 1-cell stage as a normal control exhibited two spots of macho-1 localization, which correspond to the CAB. The CAB is formed bilaterally in the posterior-most blastomeres of vegetal hemisphere, indicating the posterior pole. (**b**–**e**,**b′**–**e′**) The CytB-treated embryos represented cell-cycle retardation and various abnormal phenotypes. They were categorized into two types. In one phenotype, two CABs were formed in two neighboring blastomeres, suggesting the establishment of anteroposterior axis (**b**–**e**). The other type included the following phenotypes showing no obvious anteroposterior axis: two CABs were formed in separated blastomeres (**b′**), elongated CAB was asymmetrically inherited in two blastomeres (**c′**), three or more CABs were formed in separated blastomeres (**d′**), and a single CAB was formed in a random position of single blastomere (**e′**). Arrowheads indicate CABs. The number of embryos represented by the image over the number of embryos examined is indicated in the right corner. Scale bars: 50 µm.

**Figure 6 jdb-10-00010-f006:**
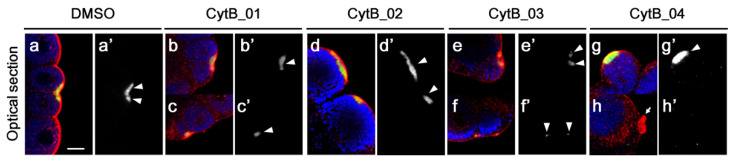
Colocalization of macho-1 with ER on the CAB at 32-cell stage. Embryos treated with CytB were fixed at 32-cell stage. The double-immunostaining of ER (red) and mitochondria (blue) and in situ hybridization of macho-1 (green) were performed. (**a**–**h**) Single optical section of merged images of CAB region. (**a′**–**h′**) Single channel of macho-1 in (**a**–**h**). Preparation name of each egg is indicated on the top. macho-1 and ER were colocalized at all the CAB regions (arrowheads) despite the CytB treatment. Only the CytB_04-treated embryo represented small debris of ER, which did not contain macho-1 (arrow). Scale bar: 10 µm.

**Figure 7 jdb-10-00010-f007:**
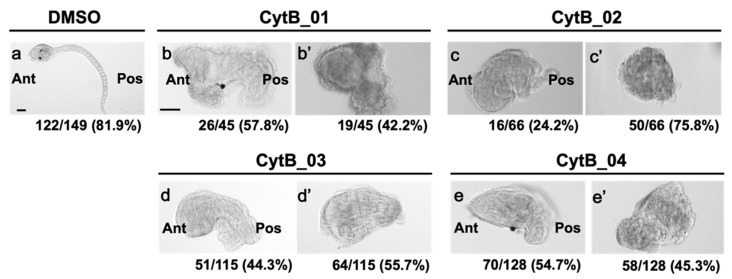
The effect of CytB treatment during the 1-cell stage on the larval morphogenesis. Preparation name of each egg is indicated on the top. Embryos were fixed at 20 h postfertilization. (**a**) The embryo treated with DMSO during 1-cell stage as a normal control exhibited tadpole shape, which has trunk and tail representing anterior (Ant)–posterior (Pos) axis. In addition, two pigment spots (otolith and ocellus) were developed. (**b**–**e**,**b′**–**e′**) The CytB-treated embryos represented various abnormal morphologies. They were categorized into two types. In one type, trunk and tail could be distinguished, suggesting the establishment of anterior–posterior axis. In some cases, otolith-like pigment granules could be observed (**b**,**e**). In the other type, embryos became cell aggregates and had no anteroposterior axis. Although they seemed to have tunic on their surface, no other obvious tissue differentiation could be observed (**b′**–**e′**). The number of embryos represented by the image over the number of embryos examined is indicated at the right corner. Scale bars: 50 µm.

## Data Availability

The datasets generated and/or analyzed during the current study are available from the corresponding author upon reasonable request.

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
