# Peer review of "Actin Filament in the First Cell Cycle Contributes to the Determination of the Anteroposterior Axis in Ascidian Development"

_jdb, 2022, doi:10.3390/jdb10010010_

Round 1
Reviewer 1 Report
Review Goto et al Journal of Developmental Biology
General comments
In the ascidian two cytoplasmic movements occur following fertilization – termed the first and second phases of ooplasmic segregation. It has been accepted for many years that the first phase depends on actin and the second phase on microtubules. This second phase of movement of organelles (ER and mitochondria domains) together with maternal RNAs leads formation of the future posterior pole of the zygote/embryo. For the first time the work presented in this article demonstrates that F-actin is also involved during the second phase of ooplasmic segregation, a process that was hitherto thought to depend solely on microtubules. The experimental data convincingly supports this claim and the analysis was appropriate.
F-actin is show to accumulate at the vegetal pole as previously found. However, for the first time it is shown that F-actin remains enriched at the vegetal pole for at least 30 min. following fertilization – importantly, up to the time that the second phase of ooplasmic segregation begins. The role of this prolonged enrichment of F-actin was evaluated by treating 1-cell stage zygotes with different duration treatment regimes of cytochalasin B. ER, myoplasm and macho 1 mRNA localization were evaluated. The convergence of the acentrosomal microtubule parallel array (CAMP) to the midline was perturbed as was the movement of mRNA macho-1 away from the ER during the second phase of ooplasmic segregation when F-actin enrichment was abolished with Cyto B.
The data presented are intriguing and convincing. My overall impression is that the article merits publication in the Journal of Developmental Biology with a few minor amendments as detailed below.
Specific recommendations
Figure 1.
Using Phalloidin staining it is shown that F-actin, which becomes enriched at the vegetal pole following the first phase of ooplasmic segregation, persists for up to 30 min following fertilization and thus up to the time that the second microtubule-dependent phase of ooplasmic segregation begins. This is thus important since the second phase of ooplasmic segregation begins when substantial enrichment of F-actin is still present at the vegetal pole.
N numbers to be added to the figure legend.
Figure 2.
To determine the role of this persistent enrichment of F-actin for the second phase of ooplasmic segregation the authors used Cytochalasin B to depolymerize F-actin. The Cyto B experiment was performed in 4 different ways –
Cyto B 1: from fertilization to 10 min. post fertilization. Relatively normal MTs.
Cyto B 2: from fertilization to 30 min. post fertilization. Perturbed MT parallel array.
Cyto B 3: from 10 min. post fertilization to 30 min. post fertilization. Relatively normal MTs.
Cyto B 4: from 30min post fertilization to 45 min. post fertilization. Missing central section of MTs.
Following each Cyto B treatment regime microtubules were stained at the beginning of the second phase of ooplasmic segregation.
Treatment regime 2 strongly perturbed the parallel array of cortical microtubules indicating that F actin accumulation to the vegetal pole during the first 10 min. following fertilization plus the persistence of F-actin at the vegetal pole from 10 to 30 min. post fertilization were both important to form the parallel array of MTs that drives the second phase of ooplasmic segregation. The data were convincing (n=55/60 for Cyto B 2 treatment regime). It was also noted that treatment with Cyto B 4 (from 30 to 45 min. post fertilization) also partially perturbed the parallel array of microtubules since the central array of MTs was absent (n=52/60). Together these data clearly indicate that the persistence of F-actin at the vegetal pole is important for the formation of the parallel array of microtubules following fertilization that drives the second phase of ooplasmic segregation.
No change to this figure suggested.
The authors should show a control Cyto B-treated zygote stained for Phalloidin to demonstrate the loss of F-actin in each treatment regime with Cyto B.
Figure 3.
The accumulation of myoplasm and mRNA macho-1 was perturbed by the Cyto B 2 treatment – in accordance with Figure 2 where this treatment regime perturbed the parallel MT array. The analysis of the data indicate that Cyto B 2 treatment regime led to less concentration of the mRNA macho-1 along the left-right plane. These data indicate that F-actin is involved during the first 10 min and during the 20 to 30 min. time window.
I would suggest that the authors also indicate the SD (as in figure 4) for parts j and k to standardize their reporting method.
Figure 4.
The Cyto B 4 treatment regime was compared to microtubule depolymerization which it thought to drive the second phase of ooplasmic segregation. It is reported that both MT depolymerization and F-actin depolymerization abolished the movement of macho-1 mRNA away from the ER that normally occurs in control embryos. This suggests that both actin and microtubules are involved in this process of mRNA delocalization from the ER.
I have no comments for this figure.
Figure 5.
The effect of Cyto B treatment during the 1st cell cycle were determined for embryonic development and CAB position at the 32-cell stage. Cyto B 2 treatment regime again had the most severe effect – unequal cell division was abolished and the CAB signal was spread over a larger area (n=45/59).
However, I was not sure c and c’ were the same stage embryos as b and b’ (these appear to be 32-cell stage), but b and b’ appear to be 16-cell stage. Please confirm the cell stage of c and c’.
Figure 6.
The CAB is affected by the Cyto B treatment regimes. And as previously reported in Fig 5, the additional function of the CAB in mediating unequal cell division is again perturbed.
Figure 7.
Cyto B treatment regimes during the first cell cycle and their effects on larval morphology is shown. Again, the Cyto B 4 treatment regime had the most severe effect on embryonic development. These data are well reported with n numbers appearing for each treatment regime.
Reviewer 2 Report
The manuscript ‘Actin filament in the first cell cycle contributes to the determination of the anteriorposterior axis in ascidian development’ by Goto et al., addresses the long standing question of how embryonic axis are established during ascidian development and in particular how determinants are partitioned during ooplasmic segregation following fertilization.
This is a fairly descriptive manuscript, which does not advance the mechanistic understanding of the process. The author revisit the role of the actin cytoskeleton during ooplasmic segregation and describe in details the changes in the localization of actin, ER, and macho-1 RNA during the first cell cycle of Ciona embryos (first cycle following fertilization), using optimized protocols for staining of fixed samples. The authors show that actin, which is known to accumulate vegetally following fertilization and during the first phase of ooplasmic segregation, remains at the vegetal side for longer than previously reported, almost until the second phase, about 30 minutes post fertilization. Using actin depolymerizing drugs, they suggest that filamentous actin is required also during the second phase of ooplasmic segregation for proper establishment of the anteroposterior axis and suggest a role for actin in organizing the CAB. Although not entirely novel, this is an interesting observation. However, there are several aspects that need to be addressed both in the analysis and the presentation.
Major points:
- The authors use cytoB treatments and wash out to test that actin has different roles at specific times of ooplasmic segregation. However, the efficiency of drug removal in wash out experiments was not evaluated. The authors should check actin organization following cytB removal.
- In Figure 2 the authors present a microtubule staining of cytB treated embryos and state that the spindles are always unaffected whereas the cortical microtubules are unaffected in treatment 1 and 3. It is actually hard to evaluate spindle organization in this pictures so an enlargement of the spindle area should be added. As for the cortical microtubules they look shorter and distributed in a wider area in cytB-03. This could be quantified.
- Based on the cytoB treatments during the first 30 minutes post fertilization the authors conclude that F-actin is required for convergence of the myoplasm to the midline. In Figure 3 b’ and d’, corresponding to the 2 brief cytoB treatments, the macho-1 RNA seems fairly concentrated (as also mentioned by the authors in line 234), differently that in the long treatment (Figure 3c’). This description does not correspond to the quantification reported in Figure 3j where the RNA appears to be spread out over a larger area. Moreover, in the images in Figure 3f-h and 3f’-h’ the macho-1 RNA seems to be significantly reduced in all 3 treatments compared to control. The amount of localized RNA could be quantified with respect to total RNA. Given the triple staining the authors should also report the effect of the treatment on the localization of cER and mitochondria at the posterior pole. Are those incorrectly localized like macho-1 RNA? From the images presented in Figure 3 the ER seems significantly reduced in treatment cytB_02 and probably in cytB_03 (hard to say from the available image). The mitochondria are less obvious and difficult to evaluate based on the images provided. In Figure 3j and k the number of embryos analyzed is very low (between 7 and 11 zygotes), and should be increased, especially given the variability observed in treated embryos. The authors should also provide some statistical analysis for the data reported.
- In Figure 4 the authors assess the role of actin and microtubules during the second phase of ooplasmic segregation. They conclude that both cytoskeletal elements have a role in macho-1 translocation. This is interesting but the effect observed in the reported pictures look very different from each other and the analysis is unclear. In CytoB_04 the ER looks very much reduced as well as the localised macho-1 RNA. Much less so in nocodazole treated embryos. Can the author quantify localized RNA? Moreover, in Figure 2 the authors show that the cytB treatment during second phase (cytoB_04) affects microtubule organization. As the second phase depends on microtubules the observed effect could be a consequence of this microtubule defect. It would also be informative to see images of the embryos at the beginning of the treatment (30 minutes post fertilization). Are the ER and the RNA completely localized and then dispersed following the treatment? Or do they fail to fully accumulate in the first place? The quantification of the signal in figure 4g is not clear. A schematic drawing showing which areas were measured could be helpful. Statistical analysis and increase of sample size are recommended.
- By analyzing the cleavage pattern of treated embryos, the authors conclude that the CAB is not properly formed in cytoB treated embryos, but both analyzed CAB markers (ER and macho1) are properly localized. Other known CAB markers could be tested to support this statement.
- The discussion is very limited and does not take into account neither what is already known in other ascidians for ooplasmic segregation nor about RNA-ER interaction in other cell types.
Minor points:
- line 60-64 a reference is missing for the work by the author showing that PEM RNAs dissociate from ER before second phase of oopmasmic segregation.
- The authors use mitochondria and MRC to refer (I think) to the same thing. They should choose one nomenclature and keep it throughout the manuscript or explain what the difference is and what are the markers used to distinguish between the two.
- In several figures, 3D models are reported, the authors should explain what they mean by that. Is it a projection of a z-stack? Or a reconstruction?
- Figure 1 legend (k, line182): the authors state that the nuclei are stained in blue, but the nucleus is never present in those stages and instead it looks like the mitochondria are stained. Is that Dapi staining? The authors should specify what markers they are using and what is stained.
- Line 268: the authors state that tubulin monomers are labelled at the vegetal pole of nocodazole treated eggs. Could they not be short nocodazole-resistent microtubules? why would tubular monomer concentrate at the vegetal pole?
- Figure 5 and Figure 7: percentages should be also added to the figure for ease of comparison.
- Figure 6: in the legend it is not indicated which markers are used and the corresponding colors used in the figure, for ER and macho-1 and for the third marker present in the merged images but never mentioned in the legend. I think it is dapi, if not necessary it could be removed.
- Figure 6 a’-h’ (legend): not clear what 'a single fluorescent channel of a merged image' is. Do they mean images of single fluorofore? Rephrase for clarity.
- Figure 4: arrowheads are used to indicate different elements, for simplicity arrows and arrowheads could be used to distinguish between macho signal and ER. Or arrowheads can be removed from panels 4 a’-b’, as they are not necessary.
Round 2
Reviewer 2 Report
The authors have addressed most of the points that were raised and the manuscript describes an interesting observation obtained with an improved methodology, both of which could be of interest to the ascidian community. I however think that some changes are still needed to improve clarity. Some are suggested below.
- 3D models should be changed to Z-projection, as there is no volume rendering.
- In legend to Figure1 chromatin should be changed for DNA as Dapi is not specific for chromatin but stains DNA in general (eg mitochondrial DNA is also stained in eggs)
-In legend to Figure 1a the authors should explain what a frontal plane is in 60mpf and how it is different from other time points or why.
- In legend to Figure 1a the authors state: ‘These observations were repeated at the four times including apploximetry 50 eggs per each.’, there are actually 7 time points in this figure, which 4 were repeated and how many times ? they should all have at least 3 replicates.
- In Figure 2b’ : the cytB treated egg seem to have a contraction pole. This is inconsistent with the role of actin in ooplasmic segregation and contraction.
- Lines 216-17: the authors state that the spindle is normally formed in cytB treated embryos, however fig 2j the spindle is abnormal and several chromosomes appear unattached to spindle microtubules. If this is not the case a more representative pictures should be selected. However a role for actin in spindle organization has already been described for starfish and, as in Figure 1 actin is observed around the spindle this result is not entirely surprising. The statement could either be rectified or removed, as it is not relevant to the rest of the analysis
- Line 263-63: ‘Effects of CytB treatments to MRC and dense ER basically showed similar localization pattern to those of macho-1and relative positions of ER, MRC, and macho-1 were conserved’. No data is shown. If the images are those in 3h-k, cytB2 looks very different from the other treatments
